# Satisfaction with Long-Term Aesthetic and 10 Years Oncologic Outcome following Risk-Reducing Mastectomy and Implant-Based Breast Reconstruction with or without Nipple Preservation

**DOI:** 10.3390/cancers14153607

**Published:** 2022-07-24

**Authors:** Rachel Louise O’Connell, Marios Konstantinos Tasoulis, Evguenia Hristova, Victoria Teoh, Ana Agusti, Ann Ward, Catherine Montgomery, Kabir Mohammed, Janet Self, Jennifer Elizabeth Rusby, Gerald Gui

**Affiliations:** 1Department of Breast Surgery, The Royal Marsden NHS Foundation Trust, Sutton SM25PT, UK; 2Institute of Cancer Research, London SM25NG, UK; marios.tasoulis@rmh.nhs.uk (M.K.T.); jennifer.rusby@rmh.nhs.uk (J.E.R.); 3Department of Breast Surgery, The Royal Marsden NHS Foundation Trust, Chelsea SW36JJ, UK; evguenia.hristova@nhs.net (E.H.); victoria.teoh1@nhs.net (V.T.); agagusti@doctors.org.uk (A.A.); ward_am@yahoo.com (A.W.); cm_m@btinternet.com (C.M.); janetself@hotmail.com (J.S.); 4Department of Research and Development, The Royal Marsden NHS Foundation Trust, Sutton SM25PT, UK; kabir.mohammed@rmh.nhs.uk

**Keywords:** breast cancer, BRCA 1 and 2, risk-reducing mastectomy, implant-based reconstruction

## Abstract

**Simple Summary:**

Women at significant increased risk of developing breast cancer may choose to undergo removal of both breasts (mastectomy) with reconstruction of the breasts using silicone implants. This study aimed to compare satisfaction, aesthetic and cancer related outcomes in women undergoing bilateral mastectomy comparing those who had the nipples removed and then a nipple reconstruction with women who had surgery with preservation of the nipples. Ninety-three women participated, sixty (64.5%) had nipple preservation and thirty-three (35.5%) nipples removed. Nipple projection was shorter in the reconstructed nipple group than the preserved nipple group. There was no significant difference in overall symmetry, satisfaction regarding nipple preservation or overall nipple satisfaction. There were no diagnoses of breast cancer in the study population who were followed up for approximately 10 years. We concluded that women who undergo nipple preserving surgery maintain long-term nipple symmetry. Nipple projection was less maintained after nipple reconstruction.

**Abstract:**

Incidence of bilateral risk-reducing mastectomies (RRMs) is increasing. The aim of this study was to compare satisfaction, aesthetic and oncological outcomes in women undergoing RRM with implant-based reconstruction comparing nipple-sparing mastectomy (NSM) with skin-sparing mastectomy (SSM) (sacrificing the nipple +/− nipple reconstruction). Women who had undergone bilateral RRM between 1997 and 2016 were invited. Aesthetic outcome and nipple symmetry were evaluated using standardized anthropometric measurements. The oncological outcome was assessed at last documented follow up. Ninety-three women (186 breasts) participated, 60 (64.5%) had NSM, 33 (35.5%) SSM. Median time between surgery and participation was 98.4 months (IQR: 61.7–133.9). Of the women, 23/33 (69.7%) who had SSM underwent nipple reconstruction. Nipple projection was shorter in the reconstructed SSM group than the maintained NSM group (*p* < 0.001). There was no significant difference in overall symmetry (*p* = 0.670), satisfaction regarding nipple preservation (*p* = 0.257) or overall nipple satisfaction (*p* = 0.074). There were no diagnoses of breast cancer at a median follow up of 129 months (IQR: 65–160.6). Women who undergo nipple-sparing RRM maintain long-term nipple symmetry. Nipple projection was less maintained after nipple reconstruction. Although satisfaction with the nipples was higher in the NSM group, this did not reach statistical significance. No breast cancers developed after RRM with long-term follow up.

## 1. Introduction

Developments in clinical genetics and improved understanding of the familial risk, has facilitated the identification of women at high risk of developing breast cancer. A greater uptake of genetic testing as well as increased awareness amongst women, for example due to the ‘Angelina Jolie effect’ [1,2], has led to an increase in requests for risk-reducing mastectomy (RRM). As a result, the number of women undergoing such procedures has been steadily rising in the UK [3], USA [1] and Asia [4].

Having a simple mastectomy can have a profound impact on emotional well-being and perception of body image. This can be mitigated to a certain extent by breast reconstruction which may improve psychological health after a mastectomy [5]. Immediate breast reconstruction was revolutionized with the introduction of skin-sparing mastectomy (SSM) by Toth and Lappert [6]. This technique allowed for the preservation of the native skin envelope while the nipple was excised. Several studies with long follow up have demonstrated the oncological safety of SSM [7,8,9,10,11,12,13]. As surgical innovation has progressed, preservation of the nipple as part of the native skin of the breast, known as nipple-sparing mastectomy (NSM), has become acceptable from an oncologic perspective [14,15,16]. Galimberti et al. [17] reviewed 1989 patients who had undergone NSM as a therapeutic procedure. After median follow up of 94 months, the incidence of local recurrence at the nipple areola complex was less than 2%.

The concept of NSM is well suited to RRM. Nipple preservation in risk-reduction surgery has been reported to be safe, although follow up in current studies is relatively short. For example, Muller et al. [18] performed a systematic reviewed of patients undergoing NSM. In the prophylactic setting there were 3716 NSM with nine cases (0.2%) of breast cancer local recurrence exterior to the nipple areola complex (NAC) and just one case (0.004%) within the NAC at an average follow up was 38.4 months (8 to 168 months). Jakub et al. [19] reviewed oncological outcomes after bilateral RRM in a population of patients with BRCA mutations. At a median follow up of 34 months, there were no episodes of breast cancer in the population.

From an aesthetic perspective, sparing the nipple preserves the identity of the breast and maintains the breast contour, potentially improving the aesthetic outcomes. However, preserving the nipple provides surgeons with the extra challenges of maintaining vascular integrity and symmetry. In case the nipple areola complex is excised, the reconstructed breast mound is flattened at the site where maximum convexity is desirable. In addition, SSM is often followed by nipple reconstruction, which as cited by women is important to make their journey complete [20]. Sparing the nipple preserves the identity of the breast and maintains the breast contour.

However, reconstructed nipples tend to lose projection over time, the tattooed nipple–areola complex often fades and, hence, the long-term satisfaction with nipple reconstruction is variable [21]. The physical impact of removing the nipple–areola complex (NAC) flattens the reconstructed breast mound at the site where maximum convexity is desirable. However, preserving the nipple provides surgeons with the extra challenges of maintaining vascular integrity and symmetry.

The primary aim of this study was to:Objectively compare nipple symmetry and projection following NSM with SSM and nipple reconstruction in women undergoing bilateral RRM with implant-based reconstruction;

The secondary endpoints were to:2.Compare participant satisfaction with the decision to preserve or sacrifice the nipples;3.Compare participant satisfaction with the appearance of the nipples;4.Compare oncological outcomes between NSM and SSM.

## 2. Materials and Methods

This was a retrospective study of women who had undergone RRM and immediate breast reconstruction with or without nipple preservation. The study was performed in accordance with the ethical standards of the institutional and national research committee (REC number 10/H0804/43) and with the 1964 Helsinki declaration and its later amendments. Data were collected and stored in accordance with the Data Protection Act (UK), the International Conference on Harmonization Guideline for Good Medical Practice and institutional standard operating procedures. Guidance from the Strengthening the Reporting of Observational Studies in Epidemiology (STROBE) [22] statement was applied.

### 2.1. Power Calculation

A symmetrical result was defined as moderate if the preserved or reconstructed nipple was within 2 cm and good if within 1 cm between the breasts for the anthropometric measurements (see Section 2.5). We estimated that a symmetrical result was less likely to be achieved in nipple sparing surgery and that approximately 50% of women having nipple sparing surgery, and 90% of women having nipple reconstruction surgery would achieve good symmetry. We assumed a response rate of 75% from amongst the estimated 90 women who had had a bilateral RRM and a 70:30 split between preserved nipples (NSM) and sacrificed (+/− reconstructed) nipples (SSM) and, therefore, required a minimum sample size of 67 women (47 in group NSM group; 20 in SSM group). These numbers would give us 90% power to detect a difference in the rate of attainment of good symmetry of 50% versus 90% (Fishers exact test; alpha 2-sided = 5%).

### 2.2. Recruitment to Study

Women aged > 18 years who had undergone RRM with immediate implant-based reconstruction to manage their high risk of developing breast cancer were invited to participate in the study. All the women underwent risk assessment with a genetics counsellor and those eligible had genetic testing for BRCA1, BRCA2 with further genetic testing as deemed necessary in keeping with contemporaneous protocols by the genetics team. Those who tested negative or did not have genetic testing, as they were not eligible at the time, were only considered for risk reduction surgery if they were high risk for breast cancer according to their risk individualised modelling using Tyrer–Cuzick, BOEDICIA or BRCA-PRO model. Women were assessed by the senior author (GG) who discussed surveillance strategies as well as risk reduction surgery in order to provide a balanced argument for both. Implant-based and autologous reconstruction using a patient-centred decision-making approach was discussed for risk management strategies and breast reconstruction type. Those who opted for an autologous reconstruction were excluded from this study. After initial surgical assessment, all women had a dedicated session with a clinical nurse specialist, the opportunity of meeting within a group counselling session, and consultation with a clinical psychologist with a special interest in risk-reducing strategies as part of the standard work-up to surgery. Women were seen for at least one further surgical consultation before surgery. Women who opted for implant-based reconstruction had specific discussion on the choice between nipple preservation or excision. Women with significant ptosis were counselled regarding the increased risk of nipple necrosis in NSM if a skin reduction technique was required and, in some instances, a SSM was recommended due to the surgeon’s concern regarding the complication risk if the patient had other risk factors. When NSM was first introduced patients were also counselled that there may be a small increased risk of development of a breast cancer compared to SSM.

The following women were excluded: those unable to complete a patient questionnaire (e.g., those with limited cognition or unable to understand and read English), women with previously treated breast cancer or those diagnosed with breast cancer leading to the decision to undergo RRM, and those diagnosed incidentally on pathological examination of the RRM specimens.

Those fulfilling the inclusion criteria were identified from a prospectively maintained database of all women who have undergone RRM under the care of the senior author between 1997 and 2016.

Women were contacted via letter with a patient information leaflet inviting them to participate in the study at their next scheduled annual follow-up clinic appointment. For those who agreed to participate during the clinic appointment, a consent form was completed. Objective measurements of nipple symmetry were undertaken during the appointment and a patient questionnaire was given to the woman to complete after the appointment and return by post. The database was updated for long-term oncological outcome at the point of last clinical follow up.

### 2.3. Operative Technique

Nipple-sparing mastectomies were performed using a peri-areolar incision with lateral extension, skin reduction (Wise) pattern or an inframammary crease incision (Figure 1). Skin-sparing mastectomies were undertaken using an elliptical incision to encompass and excise the NAC (Figure 2). Between 1997 and 2010 implants were placed in the sub-muscular position with a permanent expandable implant with complete muscle coverage (Natrelle^®^ style 150 (Allergan PLC, Madison, NJ, USA) or Siltex ™ Round Becker ™ 25 (Mentor Worldwide LLC, Johnson & Johnson Medical Limited, Wokingham, Surrey, UK)). Tissue expanders were used when patients desired breast reconstructions significantly larger than the natural skin envelopes (BioDIMENSIONAL™ McGhan 1-Stage (Allergan PLC, Madison, NJ, USA) and CPX ™ (Mentor Worldwide LLC, Johnson & Johnson Medical Limited, Wokingham, Surrey, UK)). Latissimus-dorsi-assisted breast reconstruction was performed in patients unsuitable for implant-alone breast reconstruction in women who opted against autologous reconstruction using DIEP flap. From 2010 onwards, the implants were placed in the sub-muscular position with lower pole support using a biological acellular dermal matrix (ADM) (SurgiMend^®^ PRS™ (Q Medical ™ Technologies Cumbria, UK). In keeping with breast reconstruction practice at the time, fixed volume implants were used with ADM support continuing the practice of a direct to definitive implant procedure (BioDIMENSIONAL™ McGhan 410 or 510 series (Allergan PLC, Madison, NJ, USA), Mentor^®^ CPG ™ (Mentor Worldwide LLC, Johnson & Johnson Medical Limited, Wokingham, Surrey, UK). Patients who underwent SSM were invited to undergo a subsequent nipple reconstruction and areolar tattoo.

### 2.4. Participant Demographics and Surgical Outcomes

Participant demographics, reason for undertaking RRM, type of mastectomy (nipple-sparing or sacrificing), type of reconstruction (sub-muscular implant or latissimus dorsi reconstruction with implant), type of nipple reconstruction as well as surgical complications were recorded. Demographics were presented as descriptive statistics using mean and standard deviation or median and IQR range, as appropriate, after testing for normality. Categorical data are presented as natural frequencies and proportions.

### 2.5. Objective Outcome of Nipple Symmetry

The following anthropometric measurements were undertaken for each breast separately, using a tape measure or calliper as appropriate, in cm, sternal notch to nipple, nipple to infra-mammary fold, midline to nipple, nipple diameter, nipple projection and transverse breast width. The objective data were summarised descriptively and compared between the two groups using two-sample *t*-test or Wilcoxon rank sum test. The absolute difference between the two nipples for sternal notch to nipple, nipple to infra-mammary fold and nipple to midline were used to categorise participants into one of three groups:Good symmetry: When all three parameters are <1 cm each;Moderate symmetry: If any one parameter is between 1 and 2 cm difference;Poor symmetry: If any one parameter is >2 cm different.

The proportion of participants in each group were compared using chi-squared/Fisher’s exact test for significant differences.

### 2.6. Participant Satisfaction Questionnaire

The questionnaire used was based on the questionnaire used by Didier et al. [23] and included questions to specifically address our hypotheses about projection, position and sensation of the NAC. Data from the questionnaire were expressed per participant and compared using the Mann–Whitney test for ordinal variables and Pearson’s chi-square/fishers exact test for dichotomous variables.

### 2.7. Oncological Outcomes

Participants’ electronic notes were reviewed to assess if study participants had subsequently been diagnosed with breast cancer or any other malignancy. The database was last updated in March 2021.

## 3. Results

In total, 93 women (186 breasts) participated in the study. Sixty (64.5%) women underwent NSM and thirty-three underwent SSM (35.5%). During this time, ten women underwent deep inferior epigastric perforator flap reconstruction and were not included in this study. The median time between surgery and participation in the aesthetic and participant satisfaction part of the study was 98.4 months (IQR 61.7–133.9). The median time between surgery and last follow up to assess oncological outcome was 129 months (IQR 65–160.6).

### 3.1. Participant Demographics and Surgical Outcomes

Participant demographics and surgical data are described in Table 1. Women in the NSM group had a lower BMI (23 kg/m^2^ vs. 24.4 kg/m^2^; *p* < 0.033) at the time of surgery. There was no significant difference in the age at time of surgery (*p* = 0.068) and the proportion or type of genetic mutation identified (*p* = 0.240). There were significantly more latissimus-dorsi-based reconstructions in the nipple excision group (*p* < 0.001), in patients who were more likely to have higher BMI and ptotic breasts, where central scars to excise the nipple also enabled adjustment of the skin pocket to match the implant without skin reduction incisions with junctions, such as the Wise pattern. There was a difference between the types of implants used; however, this did not reach significance. There was a preference for permanent expander-based implants if the nipples were preserved. This reflects contemporaneous practice at the time of study recruitment when total muscle coverage was used in the context of NSM with a permanent expander implant.

Of the 120 NSM (60 participants), there were 11 (9.2%) episodes of partial nipple necrosis. There were six nipples that underwent full necrosis (5%), and this resulted in two implant losses and one return to theatre for bilateral nipple excisions.

None of the women in the NSM group underwent surgery to correct nipple malposition. Of the 33 women who underwent SSM with nipple excision, 23 (69.7%) subsequently went on to have nipple reconstructions. In half the cases an arrow (Ghent) pattern, and in the other half the C-V incision was used.

### 3.2. Objective Outcome of Nipple Symmetry

Measurements of women who underwent NSM (without excision for necrosis) and SSM with subsequent nipple reconstruction are summarised in Table 2. Nipple projection was greater in the NSM group (*p* < 0.001). There was an increased sternal notch to nipple distance (*p* < 0.001) and wider nipple diameter in the SSM group (*p* = 0.003).

The aesthetic outcomes of women were assigned into good, moderate and poor symmetry groups. The majority of women in both groups were classified as having good symmetry. Only five women overall had asymmetry of more than 2 cm in any of the parameters which was the prespecified threshold for the purposes of power calculation. There was no significant difference in overall symmetry between the NSM and nipple reconstruction groups (Table 3).

### 3.3. Participant Satisfaction Questionnaire

A total of 91 participants returned their completed questionnaires, 58 were from the NSM group and 33 from the SSM group. Of the two NSM participants that did not reply, one had no documented nipple complications and the other had partial nipple necrosis on one side.

There was no difference between the NSM and SSM groups in terms of satisfaction with their decision regarding nipple preservation and overall satisfaction with the nipples. Predictably, the nipple position of surgically reconstructed nipples favoured the SSM group and sensation in the preserved nipple was superior in favour of the NSM group (Table 4).

### 3.4. Oncological Outcome

In March 2021, the electronic patient records for all participants who had undergone RRM were reviewed and the database updated to the point of censor. The median time from surgery to oncological follow up was 129 (IQR 65–160.6) months. There were no cases of breast cancer, either local or distant, in the cohort. There was one case of breast implant associated anaplastic large cell lymphoma (BIA-ALCL) in a woman with a recurrent peri-implant effusion diagnosed eleven years after breast reconstruction with textured implants.

In this cohort 59 women were diagnosed with a BRCA1 or BRCA2 mutation. Forty-five underwent bilateral salpingo-oophrectomy (60%). There were two cases of ovarian cancer in a patient with Li-Fraumeni syndrome and in a BRCA1 mutation carrier, respectively. One patient developed cervical cancer, and another was diagnosed with metastatic renal cell tumour, both had a BRCA2 mutation.

## 4. Discussion

Preservation of the NAC in NSM has gained popularity and several studies have demonstrated its oncological safety in risk-reduction surgery [19,24,25,26] though the follow-up time is relatively short in these publications. The aim of this study was to focus on the aesthetic and participant satisfaction as well as long-term oncological outcome following bilateral RRM with and without nipple preservation and nipple reconstruction.

The results of this study showed that women who undergo NSM have aesthetically acceptable results in terms of nipple symmetry and better long-term nipple projection than those undergoing SSM with nipple reconstruction. Participant-reported satisfaction regarding decisions about nipple preservation and overall aesthetic satisfaction was similar in both SSM and NSM groups. In addition, some form of meaningful nipple sensation is preserved in around half of women undergoing NSM. There were no diagnoses of breast cancer in either group after a median follow up of more than 10 years.

Women opting for RRM are usually fit and healthy, and unlike women dealing with a current diagnosis of cancer necessitating breast surgery, will not need adjuvant systemic treatment nor radiotherapy unless cancer is incidentally identified in the surgical histopathology. They are also usually young and slim and thus often well-suited to implant-based breast reconstruction. Previous studies have demonstrated that these women may have reduced satisfaction with body image [27,28] and even regret their decision for surgery [29,30]. In our experience, these women often have high aesthetic ideals and subjective comparison with their healthy breasts’ pre-surgery, meaning that the bar is set high for aesthetic outcomes; thus, expectations need to be appropriately managed to achieve long-term satisfaction.

### 4.1. Objective Symmetry

Symmetric anatomical NAC positioning is a principal goal in NSM. Choi et al. [31] reviewed a series of 1037 NSMs of which 77 (7.4%) required revision surgery for NAC position. In studies on immediate breast reconstruction for cancer, previous radiation therapy, vertical radial mastectomy incisions and autologous reconstruction were positive independent predictors of requirement for NAC repositioning, whereas implant-based reconstruction was a negative predictor for the need for repositioning. Gahm et al. [32] measured breast symmetry in women who had undergone bilateral implant-based reconstruction with subsequent NAC reconstruction by calculating the ratio between the left and right breast from the jugulum (suprasternal notch) to the reconstructed nipple, and for the midline to the reconstructed nipple and compared to a control group of women who had not undergone surgery; no significant difference in the measured asymmetry between the two groups was identified. This would be expected, since there is natural asymmetry of unoperated breasts, and when undertaking a nipple reconstruction, the surgeon and patient can plan the reconstruction in the optimal location. Therefore, it would be expected that the reconstructed NACs to be at least as symmetrical as unoperated breasts. In our study no participants in the NSM group required surgery to revise the position of the NAC, and none of the participants had the aforementioned risk factors for malposition. In our view, the greatest challenge to optimum final nipple position is degree of ptosis at the time of primary surgery and tissue elasticity over time.

We used anthropometric measurements of symmetry and showed no statistical difference in nipple symmetry between the two groups. Only five participants met the definition set for poor symmetry. Nipple projection was maintained after NSM and with long-term follow up remained greater than nipple projection after nipple reconstruction. To our knowledge, this is the first study to compare nipple projection between women who have undergone NSM and those who have undergone SSM with nipple reconstruction.

### 4.2. Participant Satisfaction

It is important to document and understand patient satisfaction and quality of life after RRM. It has previously been reported that sexual well-being and somatosensory function are most negatively affected [33]. Didier et al. [23] developed a questionnaire for women who had undergone NSM and SSM to specifically investigate the influence of nipple preservation on women who had undergone mastectomy with immediate breast reconstruction. Over 250 women completed the questionnaire which demonstrated that body image and satisfaction with appearance of the NAC were in favour of the NSM group. In a smaller study of 45 women undergoing prophylactic surgery [34], the women in the SSM group had higher satisfaction compared to the nipple preservation group according to the BREAST-Q with a mean score for ‘Satisfaction with breasts’ of 66.2 (95% CI 59–73.4) in the SSM group compared with 56.6 (95% CI 51.6–61.6) in the NSM group (*p* = 0.06). However, it is important to note that at the time of the publication, there was no specific domain in the BREAST-Q questionnaire to measure satisfaction with the nipple in case of NSM, but only measured satisfaction with the reconstructed nipple after SSM. The BREAST-Q has been revised, and since 2019 a domain for satisfaction with the nipple after NSM has been included [35]. A recent study of women who undergoing SSM or NSM mainly for oncological rather than prophylactic indications also demonstrated that there was no significant difference in satisfaction with the breasts using BREAST-Q between the two groups [36].

In our study, although a higher proportion of participants were satisfied with their NAC in the NSM group, this did not reach significance (*p* = 0.176). As expected intuitively, the women in the SSM group were more satisfied with the position of the nipples compared to the NSM group, most likely because the nipple reconstruction can be placed at the optimal position as a planned secondary procedure after the initial reconstructive surgery. Both groups of women were content after long-term follow up with the decisions made at the time of surgery to preserve the nipple or not. This is likely a reflection of shared decision making between the patient and surgeon, involving a multidisciplinary team including nurses and psychologists, so that correct individualised decisions were made by the women based on their own needs and accepted levels of risk. This decision-making process should also consider the planned reconstructed breast form, body habitus, soft tissue variables and the risk of nipple complications.

Diminished sensation of the NAC is common after NSM. Innervation of the nipple is predominantly through the anterior division of the 4th lateral intercostal nerve. We have previously reported [37] that 47% of women retained normal touch sensation of the NAC. There was some overlap of participants between the cohort of participants in the two studies. A detailed study by Benediktsson et al. [38] assessing tactile threshold of the NAC demonstrated complete loss of sensation in only 14% and normal thresholds in 31%. Petit et al. [39] demonstrated that sensitivity can increase over time. Recently, Pusic and her team have developed a new BREAST-Q module to assess sensation after breast reconstruction [40] which will be an important addition to the mastectomy and reconstruction BREAST-Q.

### 4.3. Nipple-Related Complications

Necrosis of the nipple can negatively affect the aesthetic outcome as well as the risk of infection to the underlying reconstruction which may lead to the devastating consequence of implant loss. In our series, six nipples underwent complete necrosis (5%), which resulted in one implant loss, one return to theatre for debridement and three for excision of the NAC. This is similar to other studies [41].

The choice of incision can affect the cosmetic outcome, technical ease and vascular viability of the NAC. In a small study of 37 breasts, Rawlani et al. [42] concluded that the peri-areolar incision resulted in a higher nipple necrosis rate when compared with the lateral or inframammary approach. Sacchini et al. [24] described the results of a multi-institutional study with 192 cases of NSM; the authors’ preferred incision was the transareolar/transnipple incision with medial and lateral extension which provided good vascularization to the areola and nipple, although the apical part of the two portions of the nipple was noted to be more susceptible to ischaemia and necrosis. All participants in our cohort underwent a peri-areolar incision with lateral extension or inframammary crease incision.

### 4.4. Oncological Outcome

Performing RRM has been demonstrated to reduce the risk of development of breast cancer in women at high risk and potentially improve survival [43]. Recent data from multicentre studies have confirmed the safety of NSM in the preventative setting but with a short follow-up interval. Jakub et al. [19] analysed 548 risk-reducing NSMs in 346 women with a proven BRCA1 or 2 mutation at nine institutions. At a median follow-up of 34 months, there were no episodes of breast cancer in the treated breasts. In the Memorial Sloan Kettering Hospital series [25] of 728 NSM, 459 were undertaken for risk reduction, while the remainder were therapeutic. In that study, the median follow up for the entire cohort was 49 months, and there were no new or recurrent local breast cancers identified, though there was one regional recurrence in a patient who had a therapeutic NSM. Our cohort has a significantly longer follow-up of over ten years with no post-surgical diagnoses of breast cancer. A recent Cochrane review [44] investigated the incidence of breast cancer after RRM and included several studies reporting 100% reduction in incidence of breast cancer following RRM [45,46,47]. Klijn et al. [48] demonstrated a significant risk reduction but not complete risk elimination, as one of 73 RRM participants developed breast cancer versus 23 of 173 non-RRM participants. Skytte et al. [49] found an annual incidence of breast cancer of 0.8% in the RRM group and 1.7% in the non-RRM group. Subsequent to the Cochrane review, a recent Brazilian study [50] of 62 women who underwent NSM had one patient who developed a new breast cancer after a mean follow up of 50 months. When counselling women for RRM, it is important not to state 100% reduction in risk, as there is a residual but low risk of developing a subsequent breast cancer.

### 4.5. Strengths and Limitations

This is a homogeneous cohort of women who have been confirmed to be at high risk of developing breast cancer by genetic testing and family history assessment; however, it is also real-world data from a high-volume breast unit. Recruitment met the prespecified sample size target, and the median follow up was eight years for the aesthetic outcome and more than ten years for oncological measures. We were able to objectively assess the symmetry of the nipple as well as women’s own perception of nipple preservation and nipple reconstruction. The limitation of the study was that we used a non-validated questionnaire to assess satisfaction with the nipples, as at the start of the study, questionnaires such as the BREAST-Q reconstruction module were not available [51]. Nevertheless, the simple questionnaire used was based on a validated one, that was practical to use and focused to interrogate specifically patients’ satisfaction with natural or reconstructed nipples. Another limitation is that recently pre-pectoral breast reconstruction has become a popular method of implant-based breast reconstruction; however, many surgeons continue to use the sub-pectoral implant placement. Further work is needed to assess long-term nipple symmetry and satisfaction in women who undergo risk-reducing mastectomy in pre-pectoral breast reconstruction. Early small cohort data are promising [52].

## 5. Conclusions

This study demonstrated that women who undergo a nipple-sparing risk-reducing mastectomy have aesthetically acceptable results in terms of nipple symmetry and better long-term nipple projection compared with nipple reconstruction. Satisfaction with nipples post-surgery was higher in the NSM group, but this did not reach significance and participants with reconstructed nipples were more satisfied with nipple position. Some sensation was retained in approximately half of the women who underwent NSM. There were no episodes of breast cancer with a median follow up of 129 months. Further work is needed to evaluate nipple symmetry after pre-pectoral implant-based reconstruction.

## Figures and Tables

**Figure 1 cancers-14-03607-f001:**
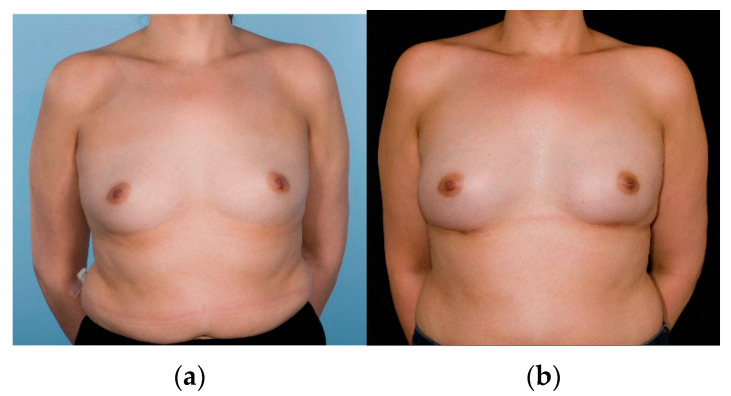
Example of a woman who underwent nipple-sparing mastectomy with implant-based reconstruction in the sub-muscular position with lower pole support using acellular dermal matrix. Pre-operative photograph (**a**) and post-operative photograph (**b**).

**Figure 2 cancers-14-03607-f002:**
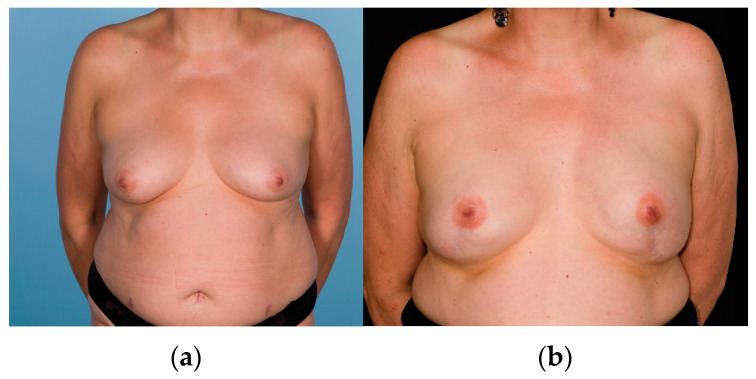
Example of a woman who underwent nipple-sacrificing mastectomy with implant-based reconstruction in the sub-muscular position with lower pole support using acellular dermal matrix. Pre-operative photograph (**a**) and post-operative photograph (**b**).

**Table 1 cancers-14-03607-t001:** Participant demographics and reconstruction type and surgical outcomes.

Items	Nipple-Sparing Mastectomy (NSM) (*n* = 60 Participants)	Nipple-Sacrificing Skin-Sparing Mastectomy (SSM) (*n* = 33 Participants)	*p*-Value
Participants demographics	Median, (IQR)	Median, (IQR)	
Time from mastectomy and reconstruction to participation in study (months)	85.5 (50.4–138.2)	116.2 (79.0–148.3)	0.136
Age at time of surgery (years)	37 (33–41)	41 (34–44)	0.068
BMI at the time of surgery (kg/m^2^)	23.0 (21.2–24.9)	24.4 (22.3–26.4)	0.033
**Participants** **genetic status**	***n* = 60, (%) participants**	***n* = 33, (%) participants**	
BRCA1	23 (38.3)	11 (33.3)	0.240
BRCA 2	14 (23.3)	11 (33.3)	
TP53	1 (1.7)	0	
Negative test results	16 (26.7)	4 (12.1)	
Unknown (not tested)	6 (10)	7 (21.1)	
**Participants** **surgery type**	***n* = 60, (%) participants**	***n* = 33, (%) participants**	
Reconstruction type			<0.001
LD + Implant	0	11 (33)	
Sub-muscular implant	60 (100)	22 (67)	
Implant used:			0.258
Tissue expander	6 (10)	4 (12)	
Permanent expander implant	44 (73)	19 (58)	
Direct to permanent fixed volume implant	10 (17)	10 (30)	
**Per breast complications**	***n* = 120, (%) breasts**	***n* = 66, (%) breasts**	
Haematoma	3 (2.5)	0	0.553
Wound infection	12(10)	4 (6.1)	0.424
Nipple necrosis partial	11 (9.1)	-	-
Nipple necrosis full thickness	5 (4.2)	-	-
Delayed nipple reconstruction			
No	-	10 (30)	-
Yes	-	23 (70)	

Latissimus dori = LD; body mass index = BMI.

**Table 2 cancers-14-03607-t002:** Anthropometric measurements to assess nipple symmetry for the left and right breast comparing the NSM and non-NSM groups. The absolute distance difference between the left and right measurements were compared between the NSM and SSM groups.

Distance Measurements for Left and Right Breasts (cm)	Nipple-Sparing Mastectomy (NSM) *n* = 114	Nipple-Sacrificing Skin-Sparing Mastectomy (SSM) with Nipple Reconstruction *n* = 46	*p*-Value
	Mean (SD)	Mean (SD)	
Sternal notch to nipple	20.73 (2.03)	22.5 (3.10)	<0.001
Nipple to infra-mammary fold	7.93 (1.62)	8.40 (1.81)	0.107
Transverse base width	13.68 (1.69)	13.95 (1.51)	0.354
Midline to nipple	10.82 (1.54)	10.97 (1.95)	0.614
Nipple diameter	1.08 (0.26)	1.26 (0.5)	0.003
Nipple projection	0.57 (0.26)	0.38 (0.23)	<0.001
**Absolute distance difference between left and right side per participant (cm)**	**Median (IQR)**	**Median (IQR)**	
Sternal notch to nipple	0.5 (0–1.0)	0.5 (0–1.0)	0.705
Nipple to infra-mammary fold	0.5 (0–1.0)	0.5 (0–1.0)	0.367
Transverse base width	0 (0–0.5)	0 (0–0.5)	0.629
Midline to nipple	0.5 (0–1.0)	0.5 (0–1.0)	0.827
Nipple diameter	0 (0–0.1)	0 (0–0.1)	0.802
Nipple projection	0 (0–0.1)	0 (0–0.1)	0.799

**Table 3 cancers-14-03607-t003:** Overall nipple symmetry comparing NSM to non-NSM.

Symmetry	Nipple-Sparing Mastectomy (NSM) *n* = 57	Nipple-Sacrificing Skin-Sparing Mastectomy (SSM) with Nipple Reconstruction *n* = 23	Total *n* = 80	*p*-Value
Good symmetry	20 (35)	6 (26)	26 (33)	0.670
Moderate symmetry	34 (60)	15 (65)	49 (61)	
Poor symmetry	3 (5)	2 (9)	5 (6)	

**Table 4 cancers-14-03607-t004:** Results of the patient questionnaire. There was no significant difference between the two groups except for nipple sensation and nipple position.

Items	Nipple-Sparing (*n* = 58)	Nipple-Sacrificing Skin-Sparing Mastectomy (SSM) (*n* = 33)	*p*-Value
	*n* (%)	*n* (%)	
Did you participate in the decision about nipple preservation or removal?			
No	0	2 (6)	
Yes	58 (100)	31 (94)	0.129
If yes, are you satisfied with the decision you made?			0.257
Very Much	50 (86)	24 (77)	
Quite a bit	6 (10)	4 (13)	
A little	0	2 (6)	
Not at all	2 (3)	1 (3)	
Did you undergo a nipple reconstruction?	-		-
No		10 (30.3)	
Yes		23 (69.7)	
	*n* = 58	*n* = 23 (Questions for participants who underwent nipple reconstruction)	
Overall, how satisfied are you with the nipples?			0.176
Very Much	41 (71)	11 (48)	
Quite a bit	14 (24)	8 (35)	
A little	2 (3)	3 (13)	
Not at all	1 (2)	1 (4)	
Is the position of the nipples the same as before the operation?			0.020
Very Much	24 (41)	18 (78)	
Quite a bit	21 (36)	3 (13)	
A little	7 (12)	2 (9)	
Not at all	6 (10)	0	
Is the projection of the nipples the same as before the operation?			0.186
Yes	34 (59)	14 (61)	
Too prominent	11 (19)	1 (4)	
Too flat	13 (22)	8 (3)	
How is the nipple sensation compared to before the operation?			<0.001
Yes, same as before	3 (5)	0	
Less than before	26 (45)	0	
A little	0	7 (30)	
None	29 (50)	16 (70)	
How would you describe the colour of the areola compared to before surgery (nipple preservation participants only)?		-	
Same	53 (89)		
Darker	0		
Lighter	6 (10)		
Does your nipple respond to cold or touch (nipple preservation participants only)?			
No	14 (24)		
Yes	45 (76)		

## Data Availability

Data supporting the reported results are available upon request from the corresponding author.

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
