# Peer review of "Satisfaction with Long-Term Aesthetic and 10 Years Oncologic Outcome following Risk-Reducing Mastectomy and Implant-Based Breast Reconstruction with or without Nipple Preservation"

_cancers, 2022, doi:10.3390/cancers14153607_

Round 1

Reviewer 1 Report

Excellent overview of an historical series on risk reducing mastectomy. It deserves publication.

Author Response

Dear Reviewer

Many thanks for your positive review to this manuscript. 

Yours sincerely

Rachel O'Connell

Reviewer 2 Report

The manuscript reports a retrospective study on a not so large cohort of high risk women on cosmetic and oncological outcome of prophylactic NSM/SSM and on its impact on quality of life and psychological-emotional consequences, with long-term follow-up. The study is interesting and analyses a current and significant issue.

Some comments:

Introduction:

The sentence “As surgical innovation has progressed, 58 preservations of the nipple as part of the native skin of the breast, known as nipple-sparing 59 mastectomy (NSM), has become acceptable from an oncologic perspective” should require specific references, given the great clinical and scientific value of the introduction of the choice of NAC conservation during mastectomy (Kissin MW, Kark AE. Nipple preservation during mastectomy. Br J Surg 1987; 74: 58–61./Petit JY, Veronesi U, Orecchia R, et al. The nipple-sparing mastectomy: early results of a feasibility study of a new application of perioperative radiotherapy (ELIOT) in the treatment of breast cancer when mastectomy is indicated. Tumori 2003; 89: 288–91).

The sentence “There are data regarding the safety of nipple-preservation in risk-reduction surgery, although follow-up in current studies is relatively short [14]” should be update and reformulated in relation to scientific evidence: the reference of Jakub et al. is certainly appropriate, even related to oncological safety of NSM in prophylactic setting, but a wider review of scientific literature would be indicated. Several studies with substantial follow up have demonstrated the oncological safety of NSM, also in therapeutic setting (Galimberti V. et al. “Oncological Outcomes of Nipple-Sparing Mastectomy: A Single-Center Experience of 1989 Patients.” Annals of surgical oncology vol. 25,13 (2018): 3849-3857. doi:10.1245/s10434-018-6759-0/Muller T. et al. “Oncological safety of nipple-sparing prophylactic mastectomy: A review of the literature on 3716 cases.” Annales de chirurgie plastique et esthetique vol. 63,3 (2018): e6-e13. doi:10.1016/j.anplas.2017.09.005).

Materials and Methods:

It would be advisable to clearly specified the clinical reason for which some women have been submitted to NAC removal (SSM). Which are the specific criteria of selection between NSM and SSM? The terms of choice and clinical conditions that motivated performing an MNS rather than SSM do not appear so clear.

Discussion:

The sentence “Diminished sensation of the NAC is common after NSM” could be better discussed citing further recent study on this topic (Tsangaris et al. Development and Psychometric Validation of the BREAST-Q Sensation Module for Women Undergoing Post-Mastectomy Breast Reconstruction Ann Surg Oncol (2021) 28:7842–7853).

Table 1.

The cohort is characterized by some women that were submitted to RRM for “Unknown (not tested)” patient genetic status (10% and 2.1% in NSM and SSM, respectively). Could Authors give a reason for the choice of risk reducing surgery in such women without a specific information on genetic risk?

Given to the consideration that the population studied included only mutation carriers individuals, not breast cancer patients, I would suggest a more appropriate definition of population studied, i.e. not “patients”, but “individuals” or “women”.

Author Response

Dear reviewer

Thank you for your review of this manuscript.

I have addressed each point in the revisions.

To summarise

  1. I have added three references regarding the oncological safety of NSM. I added the Kissin and Petit reference and I also identified a third appropriate reference by Blanckaert et all Acta Chir Belg 2021.
  2.  I have added the Galimberti reference regarding NSM in the therapeutic setting and also added the Muller reference regarding the prophylactic setting. I have summarised in more detail the Jakub data.
  3. Further details regarding patient and surgeon decision making regarding NSM/SSM are expanded upon in the methods (line 138-143)
  4. Thank you for identifying the BREAST-Q sensation paper, I have referenced this.
  5. There was a cohort of women who were not eligible for genetic testing according to the criteria at the time. These women were only considered for risk reduction surgery if they were high risk for breast cancer according to their individual risk modelling using Tyrer Cuzick or BOEDICIA modelling. I have now updated the methods to reflect this (lines 127-130)
  6. Thank you I have referred to the participants in the study as women or participants rather than patients. In some cases I have kept patients where most appropriate
Thank you again for your comments, I believe that they have strengthened the paper.  Yours sincerely Rachel O'Connell

Reviewer 3 Report

This excellent retrospective review adds to the body of literature around the oncologic safety of NSM / SSM. It also serves to add value to the perception that nipple-preservation at the time of mastectomy has advantages with regards to nipple projection and patient satisfaction. The authors should be congratulated on a good study.

Author Response

Dear reviewer

Thank you for your positive review.

Yours sincerely

Rachel O'Connell

Round 2

Reviewer 2 Report

Dear Authors, thank you very much for adhering to my suggestions, the overall quality of your interesting study has certainly improved.

Kind regards